# WHAT CAN YOU LEARN FROM YOUR MUSCLES? LEARNING VISUAL REPRESENTATION FROM HUMAN INTERACTIONS

**Kiana Ehsani**[1,2]   **Daniel Gordon**[1]   **Thomas Nguyen**[1]   **Roozbeh Mottaghi**[1,2]   **Ali Farhadi**[1]
[1] University of Washington, [2] Allen Institute for AI

## ABSTRACT

Learning effective representations of visual data that generalize to a variety of downstream tasks has been a long quest for computer vision. Most representation learning approaches rely solely on visual data such as images or videos. In this paper, we explore a novel approach, where we use human interaction and attention cues to investigate whether we can learn better representations compared to visual-only representations. For this study, we collect a dataset of human interactions capturing body part movements and gaze in their daily lives. Our experiments show that our "muscly-supervised" representation that encodes interaction and attention cues outperforms a visual-only state-of-the-art method MoCo (He et al., 2020), on a variety of target tasks: scene classification (semantic), action recognition (temporal), depth estimation (geometric), dynamics prediction (physics) and walkable surface estimation (affordance). Our code and dataset are available at: https://github.com/ehsanik/muscleTorch.

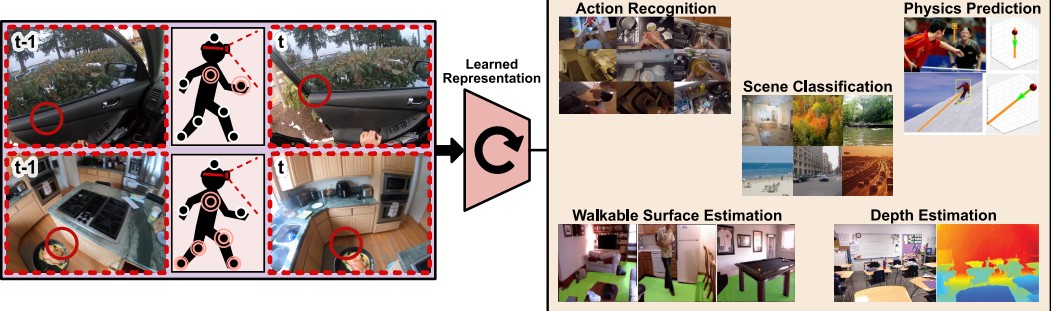

Figure 1: We propose to use human's interactions with their visual surrounding as a training signal for representation learning. We record first person observations as well as the movements and gaze of people living their daily routines and use these cues to learn a visual embedding. We use the learned representation on a variety of diverse tasks and show consistent improvements compared to state-of-the-art self-supervised vision-only techniques.

## 1 INTRODUCTION

Encoding visual information from pixel space to a lower-dimensional vector is the core element of most modern deep learning-based solutions to computer vision. A rich set of algorithms and architectures have been developed to enable learning these encodings. A common practice in computer vision is to explicitly train the networks to map visual inputs to a curated label space. For example, a neural network is pre-trained using a large-scale annotated classification dataset (Deng et al., 2009; Krasin et al., 2017) and the entire network or part of it is fine-tuned to a new target task (Goyal et al., 2019; Zamir et al., 2018).

In recent years, weakly supervised and self-supervised representation learning approaches (e.g., Mahajan et al. (2018); He et al. (2020); Chen et al. (2020a)) have been proposed to mitigate the need for supervision. The most successful ones are contrastive learning-based approaches such as (Chen et al., 2020c;b) and they have shown remarkable results on target tasks such as image

classification and object detection. Despite their success, there are two primary caveats: (1) These self-supervised methods are still trained on ImageNet or similar datasets, which are fairly cleaned up and/or include a pre-specified set of object categories. (2) This method of training is a *passive* approach in that it does not encode interactions. On the contrary, for humans, a vast majority of our visual understanding is shaped by our interactions and our observations of others interacting with their environments. We are not limited to learning from visual cues alone, and there are various other supervisory signals such as body movements and attention cues available to us. It is shown that by learning how to move the joints to walk and crawl, infants can significantly enhance their perception and cognition (Adolph & Robinson, 2015). Moreover, by observing another person interact with the environment humans obtain a visual and physical perception of the world (Bandura, 1977).

The question we investigate in this paper is, "can we learn a rich generalizable visual representation by encoding human interactions into our visual features?". In this work, we consider the movement of human body parts and the center of attention (gaze) as an indicator of their interactions with the environment and propose an approach for incorporating interaction information into the muscly-supervisedrepresentation learning process.

To study what we can learn from interaction, we attach sensors to humans' limbs and see how they react to visual events in their daily lives. More specifically, we record the movements of the body parts by Inertial Movement Units (IMUs) and also the gaze to monitor the center of attention. We introduce a new dataset of more than 4,500 minutes of interaction by 35 participants engaging in everyday scenarios with their corresponding body part movements and center of attention. There are no constraints on the actions, and no manual annotations or labels are provided.

Our experiments show that the representation we learn by predicting gaze and body movements in addition to the visual cues outperforms the visual-only baseline on a diverse set of target tasks (Figure 1): semantic (scene classification), temporal (action recognition), geometric (depth estimation), physics (dynamics prediction) and affordance-based (walkable surface estimation). This shows that movement and gaze information can help to learn a more informative representation compared to a visual-only model.

## 2 RELATED WORK

Visual representations can be learned using many different techniques from full supervision to no supervision at all. We outline the most common paradigms of representation learning, namely supervised, self-supervised, and interaction-based representation learning.

**Supervised Representation Learning.** Supervised representation learning in computer vision is typically performed by pre-training neural networks on large-scale datasets with full supervision (e.g., ImageNet (Deng et al., 2009)) or weak supervision (e.g., Instagram-1B (Mahajan et al., 2018)). These models are fine-tuned for a variety of tasks including object detection (Girshick et al., 2014; Ren et al., 2015), semantic segmentation (Shelhamer et al., 2015; Chen et al., 2017), and visual question answering (Agrawal et al., 2015a; Hudson & Manning, 2019). However, collecting a manually annotated large-scale dataset such as ImageNet requires extensive resources in terms of cost and time. In contrast, in this paper, we only use human interaction data, which does not require any manual annotation.

**Self-supervised Representation Learning.** There has been a wide range of research on self-supervised learning of visual representations in which properties of the images themselves act as supervision. The objectives for these methods cover a variety of tasks such as solving jigsaw puzzles (Noroozi & Favaro, 2016), colorizing grayscale images (Zhang et al., 2016), learning to count (Noroozi et al., 2017), predicting context (Doersch et al., 2015), inpainting (Pathak et al., 2016), adversarial training (Donahue et al., 2017) and predicting image rotations (Gidaris et al., 2018). This type of representation learning is not limited to learning from single frames. Agrawal et al. (2015b) and Jayaraman & Grauman (2015) both use egomotion, Wang & Gupta (2015) cyclically track patches in videos, Pathak et al. (2017) use low-level non-semantic motion-based cues, and Vondrick et al. (2016) predict the representation of future frames.

Inspired by contrastive learning (Hadsell et al., 2006), recent methods have used "instance discrimination" in which the network uniquely identifies each image. A network is trained to produce a non-linear mapping that projects multiple variations of an image closer to each other than to all other

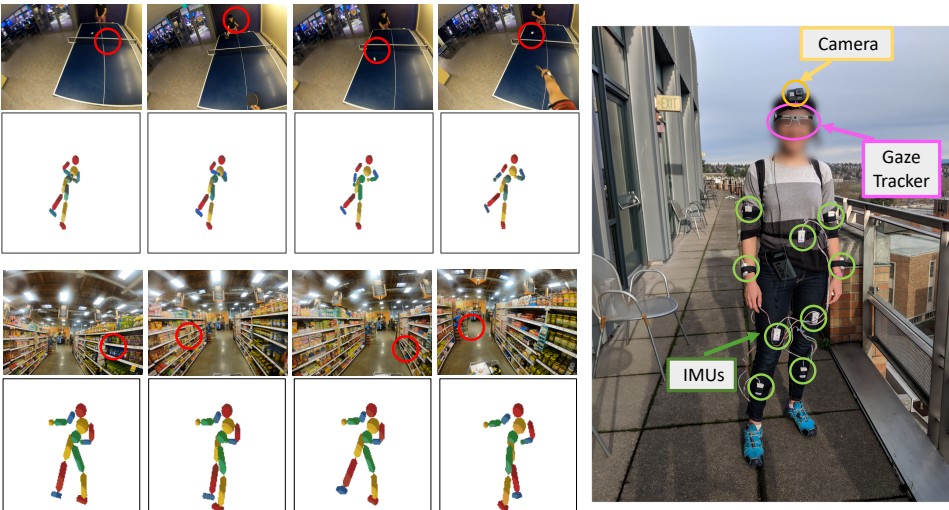

Figure 2: **Dataset examples.** Two sequences from our dataset are shown on the left. The first row shows the sequence of the images and the second row shows the movements of the body parts according to the IMU readings. We visualize the gaze using the red circle. This is just for visualization purposes and does not exist in the image. On the right, we show the data collection setup.

images. Using Noise Contrastive Estimation (Gutmann & Hyvärinen, 2010), networks are trained to differentiate between similar images under complex noise models (such as non-overlapping crops and heavy color jittering) and dissimilar images. Oord et al. (2018) and Hénaff et al. (2019) introduce and investigate the Contrastive Predictive Coding (CPC) method, which encodes the shared information between different crops of an image to predict the features from masked regions of the image. Wu et al. (2018); Misra & van der Maaten (2020) use a memory bank, which enables contrasting features of the current image against a large set of negative samples, increasing the likelihood of finding a nearby negative. The MoCo technique (He et al., 2020; Chen et al., 2020c) encodes the positive samples with a momentum encoder to avoid the rapid changes in the original feature extractor. They achieve comparable results with supervised learning representations. Chen et al. (2020a;b) show that by using a trainable non-linear transformation between the representation and contrastive latent space and larger batch sizes, they can omit memory banks entirely, allowing for full backpropagation through both positive and negative samples, and achieve better results. Bachman et al. (2019); Tian et al. (2019) maximize the mutual information between different extracted features of the same image from multiple views. Zhuang et al. (2019) enforce the extracted features of similar images to move towards the same part of the embedding space. Gordon et al. (2020), Yao et al. (2020), and Devon Hjelm & Bachman (2020) apply contrastive method to videos and leverage spatio-temporal cues to learn visual representations. Similar to these approaches, we do not rely on any manual annotation, but in contrast to them, we utilize human interactions along with their visual observation for representation learning.

**Interaction-Based Representation Learning.** The third class of learning representations relies on cues obtained by interacting with a dynamic environment. Pinto et al. (2016) learn a representation from interactions of a robotic arm (e.g., grasping and pushing) with different objects. Chen et al. (2019) and Weihs et al. (2019) both tackle the representation learning problem by training an agent to play a game in an interactive environment. Ehsani et al. (2018) learn a representation by modeling the non-semantic movements of a dog. Our work falls in this category since we use human interactions for learning the representation. We differ from these approaches in that we use low-level observations of human interaction such as body part movements and gaze to show significant improvement over a state-of-the-art baseline across multiple low-level and high-level target tasks.

## 3   HUMAN INTERACTION DATASET

We introduce a new dataset of human interactions for our representation learning framework. In this section, we describe the data collection. Our goal is to capture how humans react to the visual

world by recording their movements and focus of attention. Previous datasets of human actions and gaze include only gaze information (Fathi et al., 2012; Xu et al., 2018), part movements from a third-person view (Ionescu et al., 2014; Hassan et al., 2019), or only action or hand labels in an ego-centric setting (Damen et al., 2018; Sigurdsson et al., 2018). In contrast, our new dataset includes ego-centric observations along with the corresponding gaze and body movement information during their daily activities ranging from walking and cycling to driving and shopping.

To collect the dataset, we record egocentric videos from a GoPro camera attached to the subjects' forehead. We simultaneously capture body movements, as well as the gaze. We use Tobii Pro2 eye-tracking to track the center of the gaze in the camera frame. We record the body part movements using BNO055 Inertial Measurement Units (IMUs) in 10 different locations (torso, neck, 2 triceps, 2 forearms, 2 thighs, and 2 legs). Figure 2 shows the data collection setup along with two clips of the captured sequences. In total, we collected 4,260 minutes of videos with their corresponding body part movement and gaze from 35 people. Unlike the common large-scale datasets used for representation learning such as ImageNet, there is no restriction on the categories observed in the images, and no manual annotation is provided. Although the supervision from movement and gaze does not come for free and still requires the participants wearing the sensors, no manual or semantic annotation is needed for acquiring this dataset. Statistical analysis of the dataset is provided in Appendix A.3. Moreover, we provide details of aligning the video with the motion sensors and synchronization of the sensors in Appendix A.2. The supplementary video shows a few examples of the video clips.

## 4 INTERACTION-BASED REPRESENTATION LEARNING

Visual representation learning is typically performed using visual cues from single images or videos (He et al., 2020; Gordon et al., 2020). Our goal in this paper is to incorporate human interactions into our representations to move beyond a purely visually-trained feature representation. Below, we describe our approach for integrating movement and gaze information in the representation learning pipeline. Intuitively, body part movements should encode the temporal changes in the image based on the underlying cause of those changes (e.g., moving legs results in walking which makes distant objects move closer). Additionally, gaze grounds the visual features with the location in the image where the person pays the most attention. This should correlate well with semantic concepts such as objects, or affordances such as walkable surfaces.

### 4.1 LEARNING FEATURES

Our goal is to learn visual representations by simultaneous learning of a visual encoding for each frame and predicting body part movements and gaze attention from the sequence of observations. Formally, given an ego-centric video as a sequence of images $V = (I_t, \ldots, I_{t+k})$, the goal is to 1) estimate the gaze $G = (G_t, \ldots, G_{t+k})$, where $G_t$ is the person's center of focus in 2D camera coordinates, and 2) predict the body part movements $P = (P_t, \ldots, P_{t+k})$, where $P_t$ is a binary vector of length equal to the number of body parts, indicating whether a part is moved at time $t$.

We optimize three objectives: (1) gaze prediction, (2) body part movement prediction, and (3) auxiliary visual prediction. The visual features obtained from a CNN backbone are combined with a sequence-to-sequence model in order to predict gaze and movement. Note that the weights of the backbone are randomly initialized, i.e. we train the model from scratch. Figure 3 shows an overview of the architecture. The objectives are jointly optimized. In the following, we explain each of them in more detail.

**Gaze:** We predict the person's focus of attention by modeling their gaze in the camera reference frame. We use the Huber loss to train the center of attention $\mathcal{L}_{attention}(\hat{G}, G)$,

$$\mathcal{L}_{attention}(\hat{G}, G) = \begin{cases} \frac{1}{2} \left\| \hat{G} - G \right\|_2^2 & |\hat{G} - G| < \delta \\ |\hat{G} - G| - \frac{1}{2} & \text{otherwise} \end{cases} \tag{1}$$

**Movement:** We find the task of predicting body part movement direction and magnitude to be highly ambiguous. For example, when walking, the visual information may not show the legs, so we cannot know how high the legs were lifted. Instead, for each body part, we predict whether it is moving

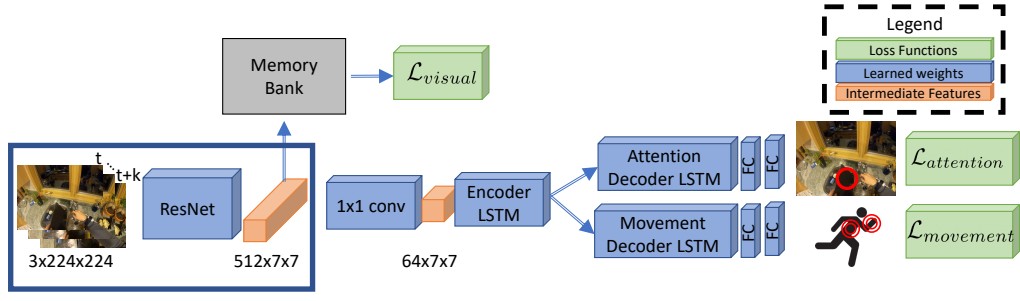

Figure 3: **Model Overview.** We learn a muscly-supervisedrepresentation by jointly optimizing visual, movement and center of focus (gaze) objectives. The portion outlined with a rectangle is the backbone that is used to evaluate the representation for target tasks. All parts of the network are initialized randomly and trained from scratch.

at all which is less ambiguous and reduces the problem to a binary classification task. Rather than predicting the movements for lower and upper parts of the joint separately (leg and thigh, forearm and tricep), we combine the movements into 6 categories of torso, neck, right arm, left arm, right leg, and left leg. To estimate this movement, we use binary cross-entropy loss and denote it as $\mathcal{L}_{movement}(\hat{P}, P)$.

**Auxiliary Visual Prediction:** We also use a visual objective, $\mathcal{L}_{visual}(\hat{I}_t, I_t)$. For this objective, similar to (He et al., 2020), we use instance discrimination. Any alternative visual encoding objective can be used instead. In section 5.3.1, we provide results for another type of visual encoding as well.

Instance discrimination's objective is to force the visual features of different augmentations of the same image to be as close as possible in the latent space (Wu et al., 2018; Chen et al., 2020a; Zhuang et al., 2019) while pushing apart all other image embeddings. By learning to extract what makes each image unique, the network focuses on semantically meaningful features of the image. This enables the feature extractor to embed a more invariant representation of the image, which is especially important when transferring to different tasks and domains. To contrast the positive samples (the augmentations of the image), with a large set of negative samples, we maintain a memory bank of embedded features from different images in the data. The final objective can be formalized as $\mathcal{L}_{visual}$ (which is also known as the InfoNCE (Oord et al., 2018) loss),

$$\mathcal{L}_{visual}(\hat{I}_t, I_t) = -\log \frac{\exp(f(I_t) \cdot f(\hat{I}_t)/\tau)}{\sum_{i=0}^{N} \exp(f(I_t) \cdot M_i/\tau)}, \tag{2}$$

where $I_t, \hat{I}_t$ are two different random augmentations of the *first* image of the sequence $V$, $f$ is the image feature extractor (ResNet backbone), $M = (M_0, \ldots, M_N)$ is the bank of negative samples and $\tau$ is a parameter that controls the concentration level of the distribution (Hinton et al., 2015). We also use a momentum-updated encoder as in (He et al., 2020). We apply the visual loss only to the first image of the sequence, as images within a sequence tend to be visually similar to each other.

The overall objective is a weighted sum of the described loss functions. More details on the architecture are provided in Appendix A.4.1.

$$\mathcal{L}_{interaction} = \alpha\mathcal{L}_{attention}(\hat{G}, G) + \beta\mathcal{L}_{movement}(\hat{P}, P) + \gamma\mathcal{L}_{visual}(\hat{I}_t, I_t) \tag{3}$$

## 4.2 ADAPTING THE REPRESENTATION TO NEW TASKS

After training the model using $\mathcal{L}_{interaction}$ objective, we use the trained weights of our feature extraction network (i.e., only the ResNet part) as the initialization for our target tasks. Our goal in this paper is to evaluate the visual representation on its own rather than using it as initialization for end-to-end training. Hence, during training for the target tasks, the weights of the feature extraction backbone are frozen. We have a diverse set of target tasks, where each requires a specific network

| Datasets | | SUN397 | Epic Kitchen | VIND | NYUv2 | |
| | | Xiao et al. (2010) | Damen et al. (2018) | Mottaghi et al. (2016a) | Nathan Silberman & Fergus (2012) | |
| Method | Training Objective | (a) Scene (Top-1 ↑) | (b) Action (Top-1 ↑) | (c) Dynamics (Top-1 ↑) | (d) Walkable (IOU ↑) | (e) Depth (RMSElog ↓) |
|---|---|---|---|---|---|---|
| MoCo (He et al., 2020) | vis | 15.80 | 24.45 | 13.18 | 58.97 | 0.148 |
| Ours | vis/attn | 21.27 | 26.80 | 13.71 | **59.65** | 0.145 |
| Ours | vis/move | 21.08 | 26.71 | 13.22 | 58.42 | **0.144** |
| Ours | vis/move/attn | **22.82** | **27.95** | **14.44** | 58.38 | 0.146 |

Table 1: **Target task results.** We compare the performance of our learned representation from movement and gaze cues with a recent self-supervised baseline MoCo (He et al., 2020) (which is trained on our data). Note that the MoCo baseline is trained using only visual data ('vis'). We evaluate the performance on a variety of different target tasks.

architecture (for example, depth estimation requires up-convolutional layers, while action recognition requires a temporal architecture). Below, we describe the result of the transfer to the target tasks. We explain the details of the architectures for each target task in Appendix A.4.2.

## 5 EXPERIMENTS

To evaluate our representation learning approach, we consider five different types of target tasks. The tasks are chosen such that they cover a wide range of domains: semantic (scene classification), temporal (action recognition), geometric (depth estimation), physical (dynamics prediction), and affordance (walkable surface estimation). We show that our learned representation, which encodes body part movement and gaze and does not rely on any manual annotation, outperforms a strong self-supervised baseline which relies on purely visual cues. Furthermore, we provide ablations of our model by using an alternative visual loss and using a subset of body parts for representation learning. For implementation details, refer to Appendix A.4.

### 5.1 SELF-SUPERVISED BASELINE

We compare our method with the recently introduced self-supervised representation learning technique, Momentum Contrast network (MoCo) (He et al., 2020), which is a state-of-the-art representation learning approach and achieves strong performance on a variety of target tasks such as image classification and object detection. The original work was trained on images from the ImageNet dataset. To ensure the comparison between our method and the baseline is fair, we train MoCo on the images from our dataset. Note that this baseline relies on visual cues only. Our goal is to show whether we can learn better representations when we use movement and gaze information in addition to the visual information.

### 5.2 EVALUATION OF THE LEARNED REPRESENTATION

We evaluate the learned representation on five different target tasks. The weights for feature extraction backbone are frozen, and only the task-specific layers are trained. We show that the representation trained using the movement and attention (gaze) supervision in addition to the visual cues outperforms MoCo (He et al., 2020) baseline (trained on our data) across the board. For each target task, we report the results in four settings, each using a different combination of visual, movement, and gaze (attention) cues for representation learning.

**Scene Classification.** For the task of scene classification, a network receives a single image as input and predicts the scene category of the image. We use SUN397 (Xiao et al., 2010) dataset for this task, as it provides a large-scale dataset of 130k images of 397 different scene categories (e.g., park, restaurant, kitchen). The results are shown in Table 1-column (a). The representation that encodes both movement and attention cues performs the best on the semantic task of scene classification. We achieve nearly a 7% improvement compared to fine-tuning the MoCo (He et al., 2020) baseline.

**Action Recognition.** The task is to predict the category of action from ego-centric videos. We use the EPIC-KITCHENS dataset (Damen et al., 2018) for this task, which is a large-scale dataset of 11M images from different action categories that are performed in various kitchens.

| Training Objective | Scene Classification Top-1 ↑ | Action Recognition Top-1 ↑ | Dynamics Prediction Top-1 ↑ | Walkable Estimation IoU ↑ | Depth Estimation RMSElog ↓ |
|---|---|---|---|---|---|
| $\mathcal{L}_{ae}$ | 11.59 | 23.84 | 9.62 | 43.64 | 0.175 |
| $\mathcal{L}_{ae} + \mathcal{L}_{att} + \mathcal{L}_{move}$ | 15.08 | 25.69 | 10.78 | 47.20 | 0.169 |
| $\mathcal{L}_{nce} + \mathcal{L}_{att} + \mathcal{L}_{move}$ | **22.82** | **27.95** | **14.44** | **58.38** | **0.146** |

Table 2: **Ablation of the visual loss.** The result of using an autoencoder for the visual loss. We re-train the models for the five target tasks. $\mathcal{L}_{att}$, $\mathcal{L}_{move}$ and $\mathcal{L}_{nce}$ are the ones used in Eq. 3.

As shown in Table 1-column (b), our method outperforms the strong baseline representation learning method by 3.5%. This again shows that incorporating additional cues such as part movements and gaze in the representation learning is beneficial for downstream tasks. It seems that both movement and attention cues are helpful for action recognition. This is aligned with our intuition that predicting the gaze of a person and how they move their body parts may be beneficial to recognizing the actions they perform.

**Future Prediction of Dynamics.** The goal of this task is to predict the future dynamics of an object in an image. We use the VIND (Mottaghi et al., 2016a) dataset for this task. It includes 150K images with corresponding object bounding boxes. The dataset categorizes physical dynamics into Newtonian scenarios such as sliding, projectile motion, and bouncing. The goal is to predict these Newtonian scenarios and the camera viewpoint for a query object that is specified by a bounding box and physical motion labels. There are 66 classes in total. The input to the network is a single RGB image and the bounding box for the query object. Table 1-column (c) includes the results for this task. We outperform the baseline by 1.3%. The representation that is learned by using both attention and movement provides the best performance for this task, which involves predicting the future trajectory of objects.

**Walkable Surface Estimation.** The goal of this task is to segment the pixels in an image that a person can walk on. We use the data from (Mottaghi et al., 2016b), which provides annotation for 1449 images of the NYU DepthV2 (Nathan Silberman & Fergus, 2012) dataset. The results are shown in Table 1-column (d). The variation of our method that uses only the gaze information achieves the highest accuracy. This might be due to the fact that, during walking, human attention is focused on the places that they can walk on. Therefore, the gaze provides sufficient information to perform this task.

**Depth Estimation.** For depth estimation, the task is to regress the values of the depth for a single monocular RGB image. We use NYU DepthV2 (Nathan Silberman & Fergus, 2012) dataset for this task, which provides 1449 densely labeled pairs of RGB and depth images. The results are shown in Table 1-column (e). Our learned representation outperforms the baseline for this task as well. Movement cues seem more aligned with the task of depth estimation, and the representation embedding with this information performs better. Note that the metrics for depth and walkable surface estimation are global metrics i.e. they are computed for the entire image. Therefore, typically a small improvement in those metrics has a significant effect on the qualitative results.

## 5.3 ABLATIVE ANALYSES

We ablate our results by replacing the InfoNCE visual loss with an autoencoder loss. Additionally, we show how gaze information affects the prediction of the body part movements. Finally, we evaluate which movements serve as an important supervision by masking out subsets of the body parts and retraining the representation.

### 5.3.1 VISUAL LOSS

As discussed in Section 4.1, we use the InfoNCE loss (Oord et al., 2018) while learning the representation. In order to investigate the impact of using this objective, we learn the representation using an autoencoder loss for our visual objective and evaluate the learned representation on all five target tasks after re-training using the new backbone. The autoencoder loss, $\mathcal{L}_{ae}$ is defined as $\mathcal{L}_{ae}(d(f(I_t)), I_t) = \|d(f(I_t)) - I_t\|_2$, where $f$ is the feature extractor backbone and $d$ is a decoder network of five up-convolution layers, which receives the $512 \times 7 \times 7$ feature as input and reconstructs a $3 \times 56 \times 56$ image.

| Prediction | Avg. Accuracy |
|---|---|
| Visual → Part Movement | 79.19 |
| Visual + Gaze → Part Movement | **81.01** |

Table 3: **Body part movement prediction.** We investigate the correlation of the movement and attention by using the human gaze as an additional input to predict the body part movements.

| Masked Parts | Scene Classification Top-1 ↑ | Action Recognition Top-1 ↑ | Dynamics Prediction Top-1 ↑ | Walkable Estimation IoU ↑ | Depth Estimation RMSElog ↓ |
|---|---|---|---|---|---|
| w/o Torso | 21.56 | 25.42 | 13.47 | 57.50 | **0.143** |
| w/o Neck | 21.50 | 26.25 | 13.54 | 56.76 | 0.148 |
| w/o Arms | 20.72 | 24.97 | 13.79 | 58.08 | 0.148 |
| w/o Legs | 21.38 | 25.65 | 12.62 | 57.16 | 0.147 |
| w/ all | **22.82** | **27.95** | **14.44** | **58.38** | 0.146 |

Table 4: **Ablation of body parts.** We show how the performance on the target tasks changes when we ignore a body part during representation learning.

Table 2 shows that gaze and movement information still provide a strong signal compared to the visual-only case. However, the results are worse than the case that we use the InfoNCE loss for learning the representation.

### 5.3.2 MOVEMENT ESTIMATION

To better understand the effect of gaze, we predict body part movements with and without gaze information. This experiment is not part of the representation learning experiments. It is just to evaluate whether using gaze provides any additional cue for prediction of the movements.

In this experiment, we predict which subset of the six groups of body parts (neck, torso, left arm, right arm, left leg, right leg) have moved. The overall architecture for this experiment is the same as our representation learning model, except for the inputs to the LSTM modules, which is instead the concatenated image features from ResNet and the embedded input gaze. The input gaze embedding is a two-layer network encoding the gaze into a feature vector of size 512.

Table 3 shows the results for this experiment. The network achieves an improvement in predicting the body parts movements by having the additional information of the person's center of attention, which can intuitively serve as a proper indicator of their "intentions".

### 5.3.3 EFFECTS OF THE BODY PARTS

To evaluate how each body part affects the learned representation, we perform an experiment where we ignore a subset of body parts, re-train the representation learning (from scratch) and evaluate the features on target tasks. Table 4 summarizes the results. The performance on the target tasks (except depth estimation) drops when we ignore a body part during representation learning. For depth estimation, removing *torso* results in a slightly lower error, which might indicate that the torso movement is not as helpful for estimating the depth.

## 6 CONCLUSION

Representations that encode movements and actions become a necessity as we move deeper towards embodied visual understanding. In this paper, we investigate the idea of using human interactions to learn visual representations. To enable this research, we introduce a new dataset of human interactions which includes hours of synchronized streams of image frames, body part movements, and gaze information across different subjects and activities. We show that representations trained to predict body movements and gaze encode additional information compared to their purely visual counterparts. More specifically, we show our representation outperforms a state-of-the-art self-supervised representation learning baseline for a variety of target tasks. Our main point in this paper

is to show there are signals other than visual information that can be used for improving visual representation. However, we require additional sensors to capture data from other modalities and we acknowledge that scalability of our dataset requires additional data collection.

ACKNOWLEDGMENTS

This work is in part supported by NSF IIS 1652052, IIS 17303166, DARPA N66001-19-2-4031, DARPA W911NF-15-1-0543 and gifts from Allen Institute for Artificial Intelligence. We thank Winson Han for helping us with making the figures for this paper.

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

APPENDIX

A.1 DATASET EXAMPLES

The supplementary video provides examples of our dataset and also qualitative results of our representation learning model.

A.2 DATA COLLECTION DETAILS

We describe the details of our hardware setup and the alignment method used to synchronize the recordings between our camera, gaze tracker, and movement sensors.

A.2.1 ALIGNMENT AND SYNCHRONIZATION OF DEVICES

There are three different devices in our setup, 1) Tobii Pro eye-tracking glasses to record gaze, 2) BNO055 IMU sensors to record movements, and 3) GoPro camera attached to the forehead to capture ego-centric videos. These different devices record data independently. Therefore, it is necessary to synchronize all recordings.

Gaze Tracker and GoPro Alignment. Tobii Pro2 eye-tracking captures a video and the center of the gaze in the camera frame. Due to the low quality of the video captured by the eye-tracking glasses, we use an additional high-quality GoPro hero 6 camera (with resolution $1920 \times 1080$ and 60 frames per second). To synchronize the videos from the gaze tracker and GoPro, we extract SIFT (Lowe, 2004) features and use a brute force algorithm for feature matching and the RANSAC method to find a homography that maps the gaze from Tobii's camera frame to GoPro's camera coordinate. Note that the gaze might be missing for some frames due to the device noise.

IMU Sensors and GoPro Synchronization. The outputs of the IMU sensors are recorded on a Raspberry Pi board. There is also a microphone on the Raspberry Pi that records the audio. We synchronize the IMU and video recordings using two methods, 1) synchronize the audio from the GoPro video and the voice recording on the Raspberry Pi board, and 2) repeat a specific pattern of body movements in front of a mirror, so it can be uniquely identified in both the movements depicted by the IMU sensors and the GoPro camera (which recorded the participant's body pose in the mirror).

A.2.2 MOVEMENT CALCULATIONS

We record the body part movements using BNO055 Inertial Measurement Units (IMUs) in 10 different locations (torso, neck, 2 triceps, 2 forearms, 2 thighs, and 2 legs). The body parts may not appear in ego-centric video frames, therefore, the task of predicting the exact orientation and location of a body part (e.g., arm) using ego-centric videos can be very challenging. We train the model using the simpler task of predicting whether a part has moved or not. This still contains rich information about the action that is happening in the video, for example, walking can be defined as periodic movements of the left and right legs.

To compute the loss function, we need to distinguish between *movement* and *no movement* in the dataset. One way is finding a threshold in the domain of the angles of the part movements, and label all the moves smaller than the threshold as *no movement* and the rest as *movement*. However, this might result in ambiguities for the movements close to the threshold, and the network might over penalize the wrong predictions in the neighborhood close to the threshold. Therefore, we add a third *gray area* label, where the network is not penalized for wrong predictions. We divide the range of movements for each sensor into three equal ranges, the first 33% is labeled as *no movement*, the last 33% is labeled as *movement* and the remaining interval is the *gray area*, for which there is no penalty for misprediction.

A.3 DATASET ANALYSIS

To ensure that the videos in the dataset consist of a wide range of activities, we do not provide any specific instructions to the subjects, and we ask them to perform their daily routine activities. Hence, the dataset includes a variety of different situations including but not limited to driving,

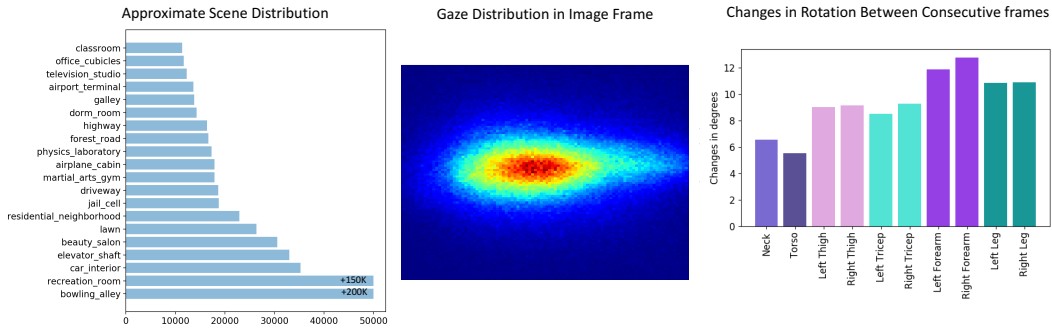

Figure 4: **Dataset Statistics.** Left: The approximate distribution of top 20 scene classes according to a scene classifier trained on the Places (Zhou et al., 2017) dataset. We show how often each scene category is predicted as top-1. Middle: The distribution of gaze across the dataset. Right: The average magnitude of the change in the orientation of body parts between consecutive frames.

cycling, playing pool, cooking, cleaning, walking in the streets, and shoveling snow. Purely for dataset analysis purposes, we gather proxy scene labels for each image. We use an off-the-shelf scene classification model trained on the Places dataset (Zhou et al., 2017) and record the top-1 prediction for each frame of our dataset. Many scene categories in our data are not present in Places, so we see a moderate amount of misclassification. However, the classifier confidently (more than 70%) predicts 101 of the 365 classes exist somewhere within our dataset, showing the diversity of our data. Figure 4 Left shows the 20 most frequent predictions. Even though there are some mispredictions among them (such as jail cell which is frequently mistaken for dark rooms), we observe that our data is fairly diverse.

Figure 4 Middle shows the distribution of the gaze in the images. As expected, the focus is mostly in the center of the image. In Figure 4 Right, we show the change in the orientation of the body parts between two consecutive frames. We observe more movements in the limbs compared to the torso and neck. We additionally notice more right arm movement than the left which is likely caused by more of our participants being right-handed.

## A.4 ARCHITECTURE & TRAINING DETAILS

In this section, we describe the details of our network architectures as well as the hyperparameters and optimization methods that we used, for reproducibility purposes. Also, the code and data will be made publicly available for further research.

### A.4.1 BACKBONE NETWORK

We train the feature extractor network by using a sequence of images of length $k = 5$, which are $\frac{1}{6}th$ of a second apart, as input. We use the ResNet18 (He et al., 2016) convolution layers as the feature extraction backbone. To preserve spatial information, which is essential for gaze prediction, we use the $512 \times 7 \times 7$ features before average pooling. We then add a $1 \times 1$ convolutional layer on top to reduce the feature size to $64 \times 7 \times 7$. The flattened feature is then input to a 3 layer LSTM with hidden size 512, which encodes the input video into a hidden feature vector. Next, the embedded video feature vector is decoded using a 3 layer LSTM, to predict the binary movement vector and gaze. For $\mathcal{L}_{visual}$, we use the feature size 128 (obtained by a fully connected layer on top of the ResNet18 features) and a memory bank of size 16384. We choose $\delta = 1$ in Equation 1, $\alpha = 0.09, \beta = 0.01, \gamma = 0.9$ in Equation 3, and $\tau = 0.07$ in Equation 2. We use a dropout of 0.5 and weight decay of 0.1. For data augmentation, we only use color jitter and random flip. We flip the entire sequence of images, swap the part movements for right and left arms and legs, and calculate the updated gaze in the flipped images.

### A.4.2 TARGET TASK NETWORKS

**Shared Implementation Details.** During the target task training, the weights for the backbone are frozen. For all of our experiments, we use the Adam optimizer (Kingma & Ba, 2015) and images

are reshaped to $224 \times 224$. The size of the hidden layer in our LSTM in all temporal experiments is 512. We use leaky-ReLU non-linearities between all network layers except for LSTMs.

**Self-supervised Baseline Details.** When training the MoCo encoder network, we use the SGD optimizer with 0.03 learning rate for the MoCo baseline since it performs best in this setting. For data augmentation, we use random cropping, horizontal flipping, gray scaling, and color jittering with the same parameters used in that work. We train the baseline with batch size 256 on 8 GPUs for 200 epochs with the same training regime as the original work.

**Scene Classification.** We use a decoder network of a single $1 \times 1$ convolution layer that reduces the feature size from $512 \times 7 \times 7$ to $64 \times 7 \times 7$, followed by two fully connected layers, that convert these features to a vector of size 512 and then 397. We use the cross-entropy loss for training, and evaluate using mean per class top-1 accuracy.

**Action Recognition.** For this task, we train a single $1 \times 1$ convolution layer that reduces the feature size from $512 \times 7 \times 7$ to $64 \times 7 \times 7$, followed by an LSTM to embed the video in one hidden vector of size 512, and two fully connected layers that convert the features to a vector of size 200 and then to the number of actions. As before, we use the cross-entropy loss as the objective and evaluate with mean per class top-1 accuracy. Since some of the action classes in this dataset appear in a limited set of videos, following one of the EPIC challenge finalists (Damen et al., 2019), we choose 9 verbs that result in a state transition, namely, *take*, *put*, *open*, *close*, *wash*, *cut*, *mix*, *pour* and *peel* and ignore verbs that do not cause a state transition (e.g., check).

**Dynamic Prediction.** We create a binary rectangular mask using the object bounding box. We use this mask image as the input to a two-layer convolutional network. As the result, we obtain a feature of size $64 \times 7 \times 7$. These two convolutional layers are trained for both our method and the baseline. We concatenate the mask feature with the feature vector obtained from the image and add two fully convolutional layers on top, to obtain the class labels. Again, we optimize the network using the cross-entropy loss and use mean per class top-1 accuracy as the evaluation metric.

**Depth Estimation.** The ResNet backbone is connected to a Feature Pyramid Network (FPN) (Lin et al., 2017). We use Pixel Shuffle layers (Shi et al., 2016) for up-scaling the lower level features. The learned ResNet backbone is frozen; the 5 up-convolution layers are the only layers trained for the target task. We use the Huber loss as the objective.

**Walkable Surface Estimation.** The architecture of this network is the same as the depth estimation network. The ResNet backbone is frozen and five up-convolution layers are the only layers that are trained for the target task. We use the binary cross-entropy loss as the objective, where the goal is segmenting walkable and non-walkable pixels. For evaluation, we use the standard Intersection over Union (IOU) metric for segmentation tasks.

## A.5 Result of Full Supervision

As a point of reference, we also provide the results using a fully supervised backbone that is trained using ImageNet. Neither our method nor the purely visual baselines use any supervision for representation learning. Therefore, a direct comparison is not fair. The results are shown in Table 5. The corresponding self-supervised results are shown in Table 1.

| Datasets | | SUN397 Xiao et al. (2010) | Epic Kitchen Damen et al. (2018) | VIND Mottaghi et al. (2016a) | NYUv2 Nathan Silberman & Fergus (2012) | |
|---|---|---|---|---|---|---|
| Method | Training Objective | (a) Scene (Top-1 ↑) | (b) Action (Top-1 ↑) | (c) Dynamics (Top-1 ↑) | (d) Walkable (IOU ↑) | (e) Depth (RMSElog ↓) |
| MoCo (He et al., 2020) | InfoNCE | 35.18 | 30.28 | 16.37 | 63.95 | 0.135 |
| Supervised | Classification | 47.27 | 32.09 | 20.01 | 65.80 | 0.132 |

Table 5: **Results of pre-training with ImageNet.** We provide the results of full supervision for pre-training using ImageNet and also the MoCo model trained on ImageNet data.

