# OpenReview forum: "What Can You Learn From Your Muscles? Learning Visual Representation from Human Interactions"
_ICLR.cc/2021/Conference — ICLR 2021 Poster_

### Official Review · AnonReviewer3 · 2020-10-25
**Where is the self-supervision signal (free) coming from?**

**Rating:** 4
**Confidence:** 5

**Review:**

The paper proposes an interesting video dataset with body-part movement signal, gaze signal. And they also treat these signals together with infoNCE/or AE as self-supervised signals. With random initialization on ResNet18, their model is better than the model only with infoNCE/AE.

I agree with the author that their supervision is working and working together with infoNCE. However, this supervision does not come for free like Rotation/Speed/Contrastive learning. For example, how do you apply your signal on the same 'egocentric' video like 'Epic-kitchen' dataset?

Pros: a new dataset contributing to the 'affordance' society, a novel and useful supervision signal.

Cons: Even though the signal is useful, it does not come for free like SpeedNet, Rotationnet or infoNCE, they still need to capture the information when recording the data.

If the author can make the supervision more free?(self-supervision), I can change my rating.

---

> ### Author Response · Authors · 2020-11-14
> **Response to R3**
>
> We thank R3 for the great feedback. We are glad that R3 found the dataset interesting and the supervision signal novel and useful.
>
> -**R3: Even though the signal is useful, it does not come for free like SpeedNet, Rotationnet or infoNCE.**
>
> We agree with R3 that our method is not as "free" as the visual-only methods such as SpeedNet, RotationNet and the others mentioned in the paper since our approach is multi-modal, and we require sensors in addition to the camera to capture data for other modalities. Our point in this paper is just to show there are signals other than visual information (such as gaze and body part movements) that can be used for improving visual representation. However, these signals might not be obtained for free. We revised our conclusion to reflect this.

---

### Official Review · AnonReviewer4 · 2020-10-28
**The paper provides a novel intuition to learn visual representations from human interactions and motion cues. The authors have well documented their research and have provided ample validation for their results, which makes this paper a notable contribution**

**Rating:** 9
**Confidence:** 4

**Review:**

The main aim of the paper is to make use of human interaction/motion to learn a visual  representation that can be re-used for classic visual tasks such as depth estimation. The authors claim that by encoding interaction and attention cues in the self-supervised representation, the method can outperform visual-only state-of-the-art methods. To study the interaction element, the authors attach sensors like Inertial Movement Units (IMUs) to the limbs of subjects and monitor their reaction to visual events in daily life. The paper also introduces a new dataset of 4260 minutes of human interactions by 35 participants which include synchronized streams of images, body part movements, and gaze information.

Paper Strengths:
This is a simply awesome paper. Idea is novel, well-validated, and well-written. The result is strong.
+ Novel intuition: The idea of the paper is intuitive, where it proposes to incorporate body part movements and gaze information in learning visual representations. Attention does play an impact in many tasks like action recognition and scene classification, which might benefit from the proposed representation learning. Also, in case of tasks like depth estimation and future prediction of dynamics, it is insightful to use body movement since it encodes temporal changes.
+ Experimental setup and ablation studies: The intuition of the authors to incorporate body part movements and gaze information in their representation has been well justified by the experimental setups and ablation studies. The importance of using each objective in the representation learning has been effectively demonstrated by showing its impact on various target tasks.
+ Performance: The authors have shown that the representation, trained using movement and attention supervision, outperforms the visual-only representations in all the tasks mentioned in the paper by a range of 1.3% to 7%
+ Dataset: The paper also introduces a new dataset of 4260 minutes of human interactions by 35 participants. This is a novel dataset with synchronized streams of images, body part movements and gaze information.

Paper Weaknesses:
- Objective function selection: In the ablation studies of body parts, we see how the removal of a body part can affect the performance on the target tasks. However, it is still not clear how each individual part may fare since the performances do not vary greatly from each other. The exclusion of the torso results in a lower error for the depth estimation, but an explanation for why exactly would that be the case may be beneficial, since intuitively it may seem that the complete movement of the body should result in better performance.

---

> ### Author Response · Authors · 2020-11-14
> **Response to R4**
>
> We thank R4 for the valuable feedback and we appreciate the positive comments about the novelty of the idea, clarity of the paper, strength of the results, novelty of the dataset, and justification of the claims.
>
> -**R4: The exclusion of the torso results in a lower error for the depth estimation, but an explanation for why exactly would that be the case may be beneficial.**
>
> This is an excellent question. We **conjecture** that information captured by torso movements for depth prediction is not as reliable as gaze or neck movements for example since torso movements intuitively do not provide much information about far regions in an image. So removing torso, makes the training easier leading to better results.

---

### Official Review · AnonReviewer1 · 2020-10-28
**Novel idea, new dataset and improvements over SOTA approach**

**Rating:** 6
**Confidence:** 5

**Review:**

This paper proposes to improve upon unsupervised representation learning for various downstream vision tasks by leveraging human motion and attention (gaze) information. The authors collect a large spatio-temporal dataset with gaze and body motion labels for this task. They train a network to jointly predict the visual focus of attention in scenes and body motion besides visual instance recognition via an NCE loss to learn good visual representations. They show large improvements in accuracy of many different visual recognition downstream tasks with their approach versus the SOTA MOCO approach, which uses visual information only.

Pros:
The work is novel and considers a new dimension to solving the problem of representation learning, which hasn't been explored before. It explores supervising neural networks to predict humans' motion and visual focus of attention. This is biological motivated by human beings' similar learning strategies. The novel datatset containing both gaze and body motion labels can be useful to the research community for other tasks beyond visual representation learning. The authors show consistent improvements with their proposed method of incorporating knowledge of gaze and body motion versus using visual information only for many downstream tasks they consider. The experimental section is fairly thorough (except for an important missing experiment as explained below).

Cons:
The authors argue that one of the disadvantages of the current approaches for unsupervised representation learning is that they use ImageNet-type datasets that require significant effort for curation and cleanup. In contrast to this, the authors' proposed approach of requiring large amounts of data with expensive TOBII eye-tracking glasses and body-IMUs from multiple subjects and with special calibration and syntonization procedures seems even more cumbersome and less accessible to ordinary practitioners of AI in the real world. How do the authors justify this? To clearly show the superiority of their dataset versus ImageNet, the authors should also include the results of MOCO trained on ImageNet for each of downstream tasks shown in their paper.
------

Post Rebuttal:
I thank the authors for their response and additional experiments to show the performance of MOCO trained on ImageNet for the various downstream tasks considered in this paper. It is evident from the results that the authors presented in Table 5 that their best method (using their multimodal data) performs worse than MOCO trained only on ImageNet with InfoNCE. Hence, while this current work has some interesting novel insights of theoretical value, I don't think the complete proposed method of data collection and training is very practical or broadly scalable. It is likely to be of limited practical applicability.  In contrast to the authors' proposed cumbersome method of collecting annotated data using expensive gaze and motion sensors, cell phones and cameras are nowadays ubiquitous and image and video data is routinely uploaded to the internet by users all over the world. Using such abundantly available existing data on the web, which can often times simply be downloaded for free without any annotations, is what I believe is likely to be a much more practical and broadly applicable approach to solving the problem of representation learning via self-supervision. This concern is also shared by Reviewer 3.

On weighing the various pros and cons of the proposed approach, I will maintain my previous rating.

---

> ### Author Response · Authors · 2020-11-14
> **Response to R1**
>
> We thank R1 for the valuable feedback. We appreciate the positive comments about the novelty of our work, the usefulness of our dataset and the thoroughness of our experiments.
>
> -**R1: How do the authors justify superiority over ImageNet?**
>
> By no means we claim superiority over ImageNet. ImageNet is an excellent, well-designed, dataset that has led to breakthroughs in supervised and unsupervised representation learning. Our point is that our dataset does not have any costly annotation compared to ImageNet which required a significant amount of money and resources for months or years of annotation, image search engines for image retrieval, etc. The cost of our sensors is negligible compared to those. Moreover our dataset is closer to the data humans use for learning (interaction with objects and scenes vs large-scale annotated datasets). However, as requested by R1, we provide the result of pre-training MoCo on ImageNet in the table below. We have added these results to Table 5 in the revised supplementary as well.
>
> |                  | Scene  Classification | Action Recognition | Dynamics Prediction | Walkable Estimation | Depth Estimation |
> | :---             |       :----:         |        :----:      |     :----:          |   :----:            |   :----:         |
> | MoCo (ImageNet)  |        35.18         |       30.28        |       16.37         |        63.95        |       0.135      |

---

### Official Review · AnonReviewer2 · 2020-10-29
**Predicting eye gaze and motion of humans improves downstream performance on visual tasks compared to visual self-supervision alone**

**Rating:** 8
**Confidence:** 4

**Review:**


# Paper Summary

The paper uses a combination of visual, human gaze and human motion sensors to build representations that perform better on downstream tasks such as action recognition, physics prediction and depth estimation than representations extracted from solely visual input. The paper announces the release a new data set of aligned visual images, eye gaze fixations and IMU motion readings from test subjects walking around an environment. Representations are computed using three different forms of information simultaneously. Given a visual input, the system tries to predict the location of eye gaze in image frame coordinates, whether each of 6 groups of motion detectors are active or not (head, torso, legs, etc.) and the result of a more traditional auxiliary visual pretext task. In this work, the paper uses “instance discrimination” where representations of augmented versions of a specific image are pushed close together in latent space and far away from augmentations of other images. Tests on diverse benchmarks show that the gaze and motion prediction improve over visual pretext tasks alone and that there is a small benefit to using both together, but it is not additive. The paper also shows the benefit of gaze and motion is present for two different visual auxiliary tasks.


# Pros and Cons

The paper defines a new way of obtaining self-supervised representations for a variety of core computer vision tasks which is important and highly relevant to the ICML community.

The paper clearly describes prior work and situates its contributions well within this space.

The paper evaluates the proposed augmented loss function on a diverse collection of standard benchmarks to test their hypothesis including scene classification, action recognition, future motion, walkable surfaces and depth estimation.

Table 1 provides a clear ablation study showing that predicting gaze and motion improve downstream performance on diverse benchmarks by non-trivial amounts:  ~7% , 3.5%, 1%, 1% and reduce RMSE  on depth estimation by 2% or so.

In the first benchmark, adding either attention or movement increases results by 6%, but adding both only improves results by 7%, not 14%! This suggests that there is a lot of redundant information between gaze and motion sensors which is a surprising as superficially they seem like very different modalities.

It is interesting and surprising that eye-gaze, where a person is looking in an image, is a useful feature for tasks such as depth prediction. This suggests there are features that are easy to compute (compared to interactions between robots and physical world)  that significantly improve network intuitions about what is important in images.

From table 4, it seems like Torso movement is more informative for scene classification, neck more informative for action recognition, arms for dynamics and arms for walkability. It is interesting that different parts are more or less informative for different tasks.

I am a little unsure what inference to draw about the results in table 2 (experiments on Lnce vs. Lae). I guess the main point is that the addition of gaze and motion also improves Lae results, so the effect is not specific to Lnce? It is interesting that gaze and motion are not as helpful with Lae ( 3% gain vs 7% with Lnce) but it is not clear why. Why would there be better complementarity between gaze, motion and NCE? Some thoughts on this would be interesting.


# Recommendation

Accept - it is an interesting and surprising result that adding self-supervised tasks for other modalities significantly improves self-supervised representations and that gaze and motion are redundant with each other.


# Questions

How are alpha, beta and gamma set? Hmm, see from appendix A.4.1 these are 0.09, 0.01 and 0.9 respectively indicating that visual loss is the primary driver here and the others are acting more like regularizations on the representation. It is a bit surprising that the other modalities have so large an effect given their small weights in the loss function. Were any other forms of regularization used such as dropout or weight decay?  Didn’t see anything about this in the appendix.

Section 5.3.2 It seems surprising that gaze, a 2 dimensional quality needs a 512 D embedding. Is this some sort of one-hot encoding over a matrix of locations?


# Feedback

Section 3: “feature extractor to embed a more detailed representation” … I don’t know if it is more detailed, but hopefully, it is more invariant, at least to the augmentations applied, ideally to all non-semantic attributes of the image.

I guess it is implied that the baseline ‘vis’ approach in the paper  is identical to the ‘vis’ in He et al 2020, but it would be nice if this was explicitly in the caption for Table 1.

---

> ### Author Response · Authors · 2020-11-14
> **Response to R2**
>
> We thank R2 for the insightful and detailed review. We appreciate the positive comments on the importance and relevance of this work, the clarity of descriptions, and diverse results and ablations. We address the questions and comments below:
>
>  -**R2: There is a lot of redundant information between gaze and motion sensors which is surprising.**
>
> A correlation between gaze and motion sensors is somewhat expected in some specific scenarios. For example, if the gaze is on a specific object, it is likely that we interact with that object. Therefore, the arm sensors will probably move as well.
>
> -**R2: What inference can we draw about the results in Table 2? I guess the main point is that the addition of gaze and motion also improves Lae results.**
>
> That’s right. Our point was to show results on a different type of visual encoder.
>
> -**R2: Why would there be better complementarity between gaze, motion and NCE compared to AE?**
>
> We conjecture that the AE provides a weaker visual signal compared to NCE, and a lot of visual information is lost. Therefore, making connections between the AE visual encoding and gaze and motion is probably much more difficult.
>
> -**R2: Section 5.3.2 It seems surprising that gaze, a 2 dimensional quality needs a 512 D embedding. Is this some sort of one-hot encoding over a matrix of locations?**
>
> The main reason for this was that the gaze embedding has a comparable size to the other features in the network. Concatenating a 2 dimensional vector with a large image embedding would probably diminish the effect of the gaze information.
>
> -**R2: How are alpha, beta and gamma set? Were any other forms of regularization used such as dropout or weight decay?**
>
> Our losses have different norms, we set these values to normalize the norm difference. We use a dropout of 0.5 and weight decay of 0.1. We have added these to the revised supplementary material.
>
> -**Feedback**
>
> Thanks for the feedback. We have applied them to the revision we submitted.

---

### Decision · Program_Chairs · 2021-01-07
**Final Decision**

**Decision:**

Accept (Poster)

**Comment:**

The paper presents an attempt to learn interaction-based representations by taking advantage of body part movements and gaze attention. Video representations are learned by benefiting from additional supervisory signals, which are not the ones commonly used, making the paper more interesting.

R3 expresses a concern that the supervisory signal does not come "for free" and that the paper is misleading. The ACs do agree with R3 that the paper benefits from additional signals and is not a pure self-supervised learning paper, strictly speaking. The authors also agreed to this in their response to the R3’s comment. R1 also mentioned (after the rebuttal phase) that the proposed approach is not a practical self-supervised learning solution and that it does not perform as effectively as conventional self-supervised learning methods like InfoNCE on Moco.

Simultaneously, the AC and the majority of the reviewers believe that the paper itself has a value as a multi-modal learning paper. We strongly suggest the authors revise the paper to remove the 'self-supervision' claim. As mentioned above, the paper is not a self-supervised learning paper and the authors are asked to correct the details of the paper to reflect this. We also recommend adding analysis on each body signal qualitatively in the final manuscript, as suggested by R4.

It will be great if the authors can consider this as a "conditional accept". In particular, the 'self-supervision' claim in the current version of the paper is misleading and this must be corrected in the final version. Note that this was also pointed out by the Program Chairs.